# Care seeking for under-five children and vaccine perceptions during the first two waves of the COVID-19 pandemic in Lagos State, Nigeria: a qualitative exploratory study

Ayobami Adebayo Bakare ![ORCID] ,[1,2] Omotayo E Olojede,[3] Carina King ![ORCID] ,[1] Hamish Graham ![ORCID] ,[3,4] Obioma Uchendu,[2,5] Tim Colbourn,[6] Adegoke G Falade,[3,7] Helle Molsted Alvesson ![ORCID] ,[1] on behalf of the INSPIRING Project Consortium

For numbered affiliations see end of article.

**Correspondence to**
Dr Ayobami Adebayo Bakare;
bakare.ayobami.adebayo@ki.se

## ABSTRACT

**Objective** To explore healthcare seeking practices for children and the context-specific direct and indirect effects of public health interventions during the first two waves of COVID-19 in Lagos State, Nigeria. We also explored decision-making around vaccine acceptance at the start of COVID-19 vaccine roll-out in Nigeria.

**Design, setting and participants** A qualitative explorative study involving 19 semistructured interviews with healthcare providers from public and private primary health facilities and 32 interviews with caregivers of under-five children in Lagos from December 2020 to March 2021. Participants were purposively selected from healthcare facilities to include community health workers, nurses and doctors, and interviews were conducted in quiet locations at facilities. A data-driven reflexive thematic analysis according to Braun and Clark was conducted.

**Findings** Two themes were developed: appropriating COVID-19 in belief systems, and ambiguity about COVID-19 preventive measures. The interpretation of COVID-19 ranged from fearful to considering it as a 'scam' or 'falsification from the government'. Underlying distrust in government fuelled COVID-19 misperceptions. Care seeking for children under five was affected, as facilities were seen as contagious places for COVID-19. Caregivers resorted to alternative care and self-management of childhood illnesses. COVID-19 vaccine hesitancy was a major concern among healthcare providers compared with community members at the time of vaccine roll-out in Lagos, Nigeria. Indirect impacts of COVID-19 lockdown included diminished household income, worsening food insecurity, mental health challenges for caregivers and reduced clinic visits for immunisation.

**Conclusion** The first wave of the COVID-19 pandemic in Lagos was associated with reductions in care seeking for children, clinic attendance for childhood immunisations and household income. Strengthening health and social support systems with context-specific interventions and correcting misinformation is crucial to building adaptive capacity for response to future pandemics.

**Trial registration number** ACTRN12621001071819.

## STRENGTHS AND LIMITATIONS OF THIS STUDY

⇒ A key strength of this study was the inclusion of perspectives from both caregivers and healthcare providers in private and public health facilities, and the recruitment of various cadres of healthcare providers.

⇒ The use of semistructured interviews, conducted while the pandemic was ongoing, provided the opportunity to understand individual perspectives and experiences.

⇒ Perspectives captured in this study may have missed some negative impacts of COVID-19 on care seeking given caregivers were recruited from health facilities and may therefore differ from the wider community.

⇒ Findings from this study may not reflect all aspects considered important to the participants as communities and healthcare workers were not consulted in the design of the interview guides.

## INTRODUCTION

The COVID-19 pandemic was declared a public health emergency of international concern in January 2020 by WHO.[1] Differential negative impacts have been reported across the globe due to the COVID-19 pandemic. While some countries have reported a high number of deaths due to COVID-19, others particularly in sub-Saharan Africa have reported low mortality,[2] but have suffered significant social and economic impacts with recovery, likely to take a protracted course.[3] As of 27 March 2022, over 8 million cases and 170 000 deaths had been reported in Africa, although estimates of actual cases (505.6 million) and deaths (439 500) in the region are much higher.[4 5] Within Africa, Nigeria reported the fourth highest number of COVID-19 cases in 2020–2021, with 215 164 reported cases

(3.4% of the African total) and 92 million estimated cases.[6] Lagos State was the epicentre of the COVID-19 pandemic in Nigeria during this period, accounting for more than 30% of Nigeria's reported cases, with the first cases identified in late February 2020.[7 8]

The pandemic has been a major stressor to health systems, exposing and exacerbating pre-existing fragility and inequities within the system.[9 10] Given the absence of effective and widely available COVID-19 treatments during the first and second waves—February to October 2020, and November 2020 to April 2021, respectively,[11] containment measures were based on public health measures like movement and travel restrictions (ie, 'lockdowns'), physical distancing, personal hygiene and use of personal protective equipment.[12] Negative impacts of these containment measures on social life and mental well-being, education, economy, health service delivery and utilisation have been reported, but mostly from non-empirical data and outside the African context.[13–18] Early predictions of Africa being hit worst by the COVID-19 pandemic did not manifest,[19] underscoring the need for context-specific empirical data. While the direct clinical impact of COVID-19 has affected adults more directly in this period, children are not exempt from indirect effects of mitigations, although observed data from Africa are lacking.[20 21]

In March 2020, the Nigerian government imposed several public health measures. The initial COVID-19 pandemic wave in Nigeria was characterised by fear, confusion and instability in the existing social structures, with misinformation fuelled by social media reports and lockdown measures imposed by the government.[7 22–25] These may have had knock-on effects on healthcare service utilisation and delivery. While multiple studies, largely from high-income contexts, have reported reductions in child illnesses and hospital admissions during periods of COVID-19 restrictions, fewer have explored the role of changes in care seeking behaviour for children during this period and their implications for future public health responses to disease outbreaks.[16 26]

In Nigeria, under-five mortality remains high, and is not on track to meet the 2030 Sustainable Development Goal global target of having less than 25 deaths per 1000 live births.[27] Pneumonia, malaria and diarrhoea are leading causes of under-five deaths in the country, responsible for almost 40% of under-five deaths in 2018.[28] Nigeria also experiences multiple outbreaks of diseases of public health significance annually, including meningococcal disease, yellow fever and Lassa fever.[29] Given the existing burden of pneumonia, malaria and diarrhoea among children, the magnitude of the COVID-19 pandemic and response, and the frequency of disease outbreaks requiring public health response which may require mass vaccination, it is important to understand how the COVID-19 pandemic affected care seeking for under-five children as well as decision-making around vaccine introduction for outbreak control. We therefore aimed to understand care seeking practices for young children

and the context-specific direct and indirect effects of public health interventions during the first two waves of COVID-19 pandemic and decision-making around vaccine acceptance at the start of COVID-19 vaccine roll-out in Lagos State, Nigeria.

## METHODS

### Study design

This was an exploratory qualitative study using reflexive thematic analysis according to Braun and Clark.[30] We conducted semistructured interviews with caregivers of children under five and healthcare providers to gather perspectives on care seeking practices during the first two waves of the COVID-19 pandemic in Lagos State, Nigeria (February to October 2020, and November 2020 to April 2021). The study was conducted as part of the process evaluation of the Lagos INSPIRING Project, which is evaluating a child pneumonia health system intervention. We followed the Consolidated Criteria for Reporting Qualitative Research guidelines for reporting.[31]

### Setting

The study was conducted in Ikorodu local government area (LGA) in Lagos State. Lagos is the most populous state in Nigeria with an estimated population of 24.6 million people in 2022,[32] and is an economic hub in West Africa. Ikorodu is one of five administrative divisions of Lagos. It is a periurban area, with fishing as the predominant economic activity in the rural parts of the LGA, and small-scale and medium-scale entrepreneurship as the major economic activity in the urban parts of the LGA. The LGA is served by 2 government-owned secondary health facilities (general hospitals), 28 primary healthcare centres (PHCs) and over 100 private facilities. Of the 28 PHCs, seven are designated as 'flagship' facilities by the Lagos State government, as they have more personnel and equipment and run 24-hour services for children and adults. There is at least one flagship PHC in each of Ikorodu's six local council development areas and all of them remained open during the first two waves of the pandemic. The flagship PHCs also acted as COVID-19 vaccination centres, except one facility which did not have a medical doctor.

As part of the public health measures, Lagos was placed on lockdown by the Federal Government of Nigeria on 30 March 2020.[7] The lockdown lasted 35 days and included a ban on social and economic activities, restriction of all non-essential movements, suspension of commuter services, closure of schools and retail shops and prohibition of mass gatherings except for funeral services.[33] Unlike PHCs and private health facilities, service provisions were limited to emergency cases in the public secondary-level facilities. A gradual easing of the lockdown commenced from 4 May 2020 with no reinstatement of movement restrictions during the second wave (see online supplemental appendix I).[7] In addition, there was a period of civil unrest in Lagos, including Ikorodu

**Table 1** Summary of participants' characteristics

| Caregivers n=32 | | Healthcare providers n=19 | |
|---|---|---|---|
| Gender | | Gender | |
| Male | 0 (0.0) | Male | 5 (26.0) |
| Female | 32 (100.0) | Female | 14 (74.0) |
| Mean age (±SD) | 31±5.0 years | Mean age (±SD) | 38±8.1 years |
| Median number of children (range) | 2 (1–5) | Median year of experience | 11 (2–40) |
| Educational level | | Educational level | |
| Primary | 2 (6.3) | Diploma | 9 (47.4) |
| Secondary | 13 (40.6) | Tertiary | 9 (47.4) |
| Tertiary | 17 (53.1) | Postgraduate | 1 (5.2) |
| Religion | | Religion | |
| Christianity | 25 (78.1) | Christianity | 15 (78.9) |
| Islam | 7 (21.9) | Islam | 4 (21.1) |
| Occupation/cadre | | Occupation/cadre | |
| Self-employed | 21 (64.5) | Doctor | 7 (36.8) |
| Employed | 5 (16.1) | Nurse | 6 (31.6) |
| No employment | 6 (19.4) | CHEW | 6 (31.6) |

CHEW, community health extension worker.

LGA (the 'EndSARS' protests against police brutality[34]), between 8 and 22 October 2020, when a curfew was imposed.

### Participants and sampling

We purposively selected healthcare providers who attended to sick children from the seven flagship PHCs and six nearby private facilities (table 1). To ensure representation of each cadre of healthcare provider, the categories of staff targeted for recruitment (nurse, community health workers and doctors) were adapted to each facility. We recruited caregivers of children under 5 years presenting at the outpatient departments (ie, with an illness) or immunisation clinics (ie, healthy children) of seven flagship PHCs and one secondary hospital. Caregivers were recruited by female clinical project staff, who screened every child brought to outpatient departments of the facilities for pneumonia. In each facility, we used convenience sampling to recruit four caregivers of under-five children at random (n=32): two caregivers of an acutely unwell child (from outpatients) and two caregivers of a child with no current illness episode (from the immunisation clinic). This sample size was based on practical considerations of the time needed to recruit participants and the expectation that it would be sufficient numbers to achieve saturation. All participants approached for the study agreed to take part.

### Data collection

Interviews were conducted from 10 December 2020 to 18 March 2021. The semistructured interview guides were based on the literature on care seeking practices and knowledge about COVID-19 during the INSPIRING Project formative phase and revised to capture the emerging COVID-19 vaccine programme roll-out in Nigeria. The interview guide for caregiver interviews had three sections focused on: participants' family and sociodemographic information, their experiences of 2020 in light of COVID-19 including their perception of the illness and economic impacts, and care seeking practices for children under 5 years. The interview guide for healthcare provider interviews had three sections focused on: service provision, facility adaptation to the COVID-19 pandemic and care seeking for sick under-five children (see online supplemental appendices II–IV).

The research team comprised paediatricians, social science and public health specialists. The interviews were conducted by OEO, a male master's student from Nigeria with experience of the local context, with support from the female clinical study staff who recruited participants based at each facility. Interviews were conducted in English or Yoruba (the indigenous local language in Ikorodu LGA), depending on the participant preference. The interviewer lived in Ikorodu before and during the COVID-19 pandemic and had previously visited the participating health facilities for other data collection activities.[35] Caregivers' interviews were conducted at the health facility or in another convenient place agreed by the participants. Providers' interviews were held at the facility. Each interview lasted between 30 and 40 min and no repeat interviews were carried out. All interviews were voice recorded, transcribed and translated into English before being stored in a secure cloud platform with access granted to only research team members. No transcripts were returned to the participants for review.

## Data analysis

After cross-checking of the transcripts, the analysis team (AAB, OEO, HMA and CK) conducted a data-driven thematic analysis to develop themes and subthemes.[36] AAB and OEO independently reviewed all the transcripts to identify initial codes which were reconciled in NVivo.[37] Healthcare provider and caregiver interviews were initially coded separately, and then reviewed by the analysis team to identify common themes and subthemes, which were refined in subsequent analysis meetings. The process continued until the patterns of meaning were clear. The unit of analysis was COVID-19-related responses in the interviews.

## Patient and public involvement

The overarching study was designed through a codesign workshop involving representatives from the Nigerian governments, community-based organisations, professionals, Save the Children and evaluation partners. However, patients were not involved in the design of this study. Findings from this study were not discussed with the participants, but will be incorporated into the final report that will be disseminated to the relevant stakeholders including healthcare providers and community-based organisations.[38]

## FINDINGS

We identified two overarching themes which were common to caregivers and healthcare workers: appropriating COVID-19 in the belief systems, and ambiguity towards preventive measures (table 2). When the findings differ between healthcare providers and caregivers, this is specifically noted in the text.

## Appropriating COVID-19 in belief systems

This first theme elucidates plurality in the placement of COVID-19 within the context of existing belief systems. Caregivers and healthcare providers ascribed various causes to the emergence of COVID-19 including political, religious, social and geographical dimensions.

From the healthcare providers' interviews, social and political placements of COVID-19 emergence were commonly reported. To some healthcare providers, COVID-19 was not perceived as a public health problem in Nigeria.

> Except that they would say that I am a medical practitioner but I still have the impression that there is no COVID in Nigeria. Don't mind me, it's just my own belief. (Doctor—male, public facility)

The COVID-19 pandemic was framed through a political lens, with distrust in the government shaping disbelief in the disease. This distrust in government provided an opening for misinformation about the virus and control measures with participants describing COVID-19 as 'a lie' and 'a deceit from the government'. The distrust also fed into caregivers' perceptions about COVID-19 surveillance, with some caregivers reportedly delaying care seeking to avoid being automatically added to the COVID-19 daily government case list. The disbelief of the existence of COVID-19 had social associations with participants believing that the disease would not affect 'the poor' or 'black man'.

> There were some people that were like nothing is happening, we've not seen someone with it here, none of our relatives had it so it's just a scam. They don't believe it, most people don't believe it. (Community health extension worker (CHEW)—female, public facility)

| Organising themes | Themes | Subthemes |
|---|---|---|
| Appropriating COVID-19 in the belief systems | Political placement of COVID-19 | Disbelief in the virus existence |
| | | Misinformation and misconceptions about COVID-19 |
| | Sociotheological placement of COVID-19 | Religious explanation for COVID-19 |
| | | Social placement of COVID-19 |
| | Medical placement of COVID-19 | COVID-19 infection is real |
| | | Healthcare as a source of infection |
| Ambiguity about COVID-19 preventive measures | Unappealing lockdown experiences and associated adaptive mechanisms | Direct impact of lockdown |
| | | Indirect impact of lockdown |
| | | Health system adaption and its consequences |
| | Drivers of COVID-19 vaccine hesitancy | Misinformation and conspiracy theories about COVID-19 vaccine |
| | | Fear and worries about COVID-19 vaccines |
| | | Distrust in government efforts regarding COVID-19 vaccines |
| | | Media influence on COVID-19 |
| | Drivers of COVID-19 vaccine uptake | Motivation to accept COVID-19 vaccine among healthcare providers |
| | | Motivation to accept COVID-19 vaccine among community members or caregivers |

**Table 2**  Summary of themes and subthemes

To some caregivers, COVID-19 was symbolic and they offered religious explanations, describing it as a test of faith, signs of the 'end of time', a 'punishment from God' or the 'work of the devil', but this was not apparent among healthcare providers.

It's just like God wanted to deliberately punish people for their bad behaviours […]. Before, when one is sick, they'll say they should carry the individual, if it's our governors, they'll take flight and fly them out of the country. But when COVID-19 came, no one can come inside or go outside. Everyone is static (immobile in lockdown), so it's not COVID-19 again. It's God's judgement on us. (Mother—sick child, one child)

Other participants believed that COVID-19 existed as a symptomatic disease caused by a medical germ. Healthcare facilities were described as 'contagious'—a source of infection, and hospital avoidance during the acute phase of the pandemic was reported by both caregivers and healthcare providers. Given health facilities were considered high-risk places, this perception resulted in (1) no care seeking practices for some sick under-five children as caregivers resorted to self-treatment of their child's illness by seeking care from drug sellers instead, and (2) delayed presentation at health facilities when the child's condition had worsened. Similarly, when caregivers identified COVID-19 signs in their child they avoided hospital for fear of COVID-19 diagnosis or referral to isolation.

They didn't come. A lot of people were practicing self-medication. People who had cough for example, they didn't come for treatment for fear of being told they had COVID. They kept managing it at home. (CHEW—female, public facility)

Like one of my neighbours when her baby was running a temperature, she could not bring the baby to the hospital because she said when she goes to the hospital - now they will say her baby have this thing high fever, they should take him to isolation center. Because of that she now went to the pharmacy and brought some (medicine). (Mother—healthy child, three children)

Both caregivers and healthcare workers reported being extra careful in hospital settings, and sometimes this led to inaccessibility of care if healthcare providers suspected COVID-19 or had inadequate protective equipment. In contrast, one healthcare provider noted that service delivery for children did not change, stating that COVID-19 infections in children are not as severe as that of adults, and it would be unethical to deny children access to healthcare.

## Ambiguity about COVID-19 preventive measures

This theme details various responses, experiences and effects of recommended COVID-19 preventive measures and associated adaptations.

The lockdown was perceived as an unpleasant and difficult period as participants were restricted to indoor livelihoods with little or no access to transportation. Caregivers reported indirect effects of lockdown that could affect care seeking, including diminished household incomes which necessitated loan acquisition or seeking help from family members. Household food insecurity was exacerbated, and caregivers reported reducing their consumption to save food for their children. There was avoidance of social functions, mental health challenges and a focus on basic needs:

l have two teachers in my compound, not government teachers but private teachers. When the lockdown started then, the man is a teacher in private school, the woman is a teacher in a private school. As the school was not open, no salary, no money, nothing, nothing. For them to feed was problem, [never mind] if the baby falls sick, and now there is no money to take the baby to hospital. Sometimes, they will go and do herbal, this thing agbo (herbal concoction). (Mother—healthy child, three children)

Health facilities made adjustments to ensure continuous service delivery without undermining safety. Face masking, physical distancing and improved personal hygiene were adopted; however, they created additional problems such as discomfort (face masking), denied access to care or seeking medical advice from people without medical training. Caregivers complied with the rule although there were reports of anger and verbal assaults on healthcare providers when these measures were enforced at the health facilities.

There was a continuation of routine vaccination services during the lockdown, but caregivers' incorrect assumption of PHC closures during the lockdown (secondary facilities were closed to non-emergency cases), compliance with the lockdown order and fear of COVID-19 partly contributed to reduced attendance at the immunisation clinic as reported by a CHEW:

If you remember even on social media (mass media), it was broadcasted that if what you want to do at the hospital is not very important, stay indoors and stay safe. So people adhered to that rule, to the extent that when we went for outreach services, we asked them why they haven't been coming for immunization. Then they will say it's because of the lockdown, and then 'corona' stopped us from coming out. They would also claim they don't know that the facility still runs its services. (CHEW—female, public facility)

When COVID-19 vaccines became available in Nigeria, there were mixed perceptions and ambiguity towards them. Among some caregivers, the vaccine was regarded as 'a mark of the beast', or a depopulation strategy from Western countries. Religious belief, misinformation and fear of side effects were reasons identified by caregivers for COVID-19 vaccine hesitancy. Healthcare providers, in contrast, expressed distrust in the government and were

concerned about vaccine safety, quality, short timeline for vaccine development and the government's aggression towards COVID-19. They believed the vaccines were not tested very well in Nigeria before being approved.

> That thing (COVID-19 vaccine) is not well tested that's my point. It's supposed to go through a series of tests before allowing it to come into this country. So I cannot even advise anyone to take it. (Nurse—female, private hospital)

Social media (WhatsApp, Facebook, Instagram) was identified as a source of misinformation about the vaccine. One healthcare provider queried the decision of the government to accept donated vaccines that are being rejected by other countries, as reported on social media. Similarly, vaccines sent to Nigeria were presumed to be of suboptimal quality compared with the ones used abroad but this was linked to distrust in governments.

> Some people (healthcare providers) don't want to take it because of the things we have seen on social media that if you take it, it can cause this and that. (CHEW—female, public facility)

However, some healthcare providers and caregivers had positive perceptions of the vaccine, describing it as beneficial to the recipients, such as preventing sudden death and protecting against the virus. Others also showed trust in the government believing that the government cannot bring vaccines if they are harmful. Some caregivers also expressed willingness to receive the vaccine given that they are using an existing routine immunisation programme.

> If the vaccine comes, we know there's a reason why the government brought it. It has a work it wants to accomplish, which is why they want to bring it; we will take it. (Mother—sick child, four children)

Perceived higher risk of infection, the possibility of vaccines becoming scarce, a sense of responsibility to clients, motivation from senior colleagues or health managers and later positive testimonies from recipients were identified as drivers of uptake among healthcare providers. Being a requirement for overseas travel or pilgrimage, counselling and public awareness were reported by healthcare providers as drivers of vaccine uptake among community members. Few healthcare providers who had taken the vaccine identified self-reflection and personal inquiry as ways they dealt with the misinformation about the vaccine.

> I heard they were cloning the vaccine in some European countries. That was my fear but when I did my own research. I found out that there is no issue. (Doctor—female, public facility)

Despite the fear and negative perceptions, community members turned out en masse to receive the vaccine, and turnout exceeded expectations, making the supply inadequate.

> We were even surprised. I wasn't expecting people to come out. It was supposed to be a 10-day program [...] but we extended further for four weeks or thereabout. People were still coming, we had to tell them that there was no more vaccination. (Doctor—male, public facility).

## DISCUSSION

It is important to understand both community and healthcare workers' perceptions and experiences during the initial COVID-19 waves to adapt the provision of healthcare services to children during future pandemics. In the Nigerian context, participants reported both direct and indirect effects on care seeking for children, especially during the acute lockdown periods. Both groups of participants interpreted the COVID-19 pandemic through medical, political, social and economic lenses; however, religious interpretation of the pandemic was more prominent among caregivers. Care seeking for children under five was affected in part due to the perception of healthcare settings being contagious, fear of COVID-19 diagnosis and limited access to transportation. Adapting to seek care from alternative sources for sick children was reported by both groups. COVID-19 vaccine hesitancy was a major issue among healthcare providers, but less so among community members at the time of vaccine roll-out in Lagos. The motivations for vaccine uptake differed between the groups, and social media seemed to play a crucial role in shaping acceptability of the COVID-19 vaccine.

Our study suggests that COVID-19-related misinformation, rooted in a general distrust of government and cutting across every aspect of the COVID-19 response (including vaccine roll-out), had negative influences on care seeking for children. This resonates with findings elsewhere in Africa and globally that misinformation and misleading interpretations of health information (eg, daily reporting of cases and deaths from COVID-19 and fear of being counted as a COVID-19 case, assumption of facility closure during the lockdown) contributed to hospital avoidance,[16 39 40] and therefore require consideration and active management in future outbreaks.[41] Conversely, the diversity in COVID-19 placement could conceivably have positive influences on care seeking. For instance, religious beliefs relating to COVID-19 may provide emotional resilience and motivate caregivers to do everything possible to protect their children.[42] Fear of COVID-19 may similarly motivate caregivers to seek care early and get vaccinated, and even a disbelief in COVID-19 may motivate caregivers to go about business as usual.

While there were people who did not believe in COVID-19 and/or did not seek care to avoid being caught up in the response (eg, wanting to avoid isolation centres), some took it seriously and many integrated religious interpretations into their understanding of the disease. A study conducted in Nigeria found that religion

and religious institutions, focused on Christianity, could have a negative influence on illness perception and behaviour, but that most Nigerian Christians comfortably integrated religious and physical health domains.[43] Additionally, some religious organisations actively encouraged adherence to COVID-19 preventive measures.[43] These findings highlight the dynamic process of classifying new diseases, as seen in the emergence of Ebola disease,[44] and the need for sociocultural considerations and community participation in public health planning and communication, as well as active feedback and management of rumours and misinformation during the response.[45 46]

When caregivers decided to seek care for their children, lack of transportation due to lockdown inhibited access. Our finding agrees with an online survey conducted in Nigeria,[47] but contrasts with a study conducted in the Netherlands which reported parental non-deterrence in care seeking for a sick child.[48] Though the nature of illness could have been responsible for this contrasting finding, given the different epidemiological profiles, differences in health systems, COVID-19-related public health measures, as well as better health literacy around COVID-19 also have modulating effects. As reported in the UK, positive experiences from the National Health Service and support from others were positive influencers of care seeking, whereas fear driven by media and community were barriers to parental care seeking.[49] Worsened household income and food security reported during the acute phase of COVID-19 are in keeping with findings in other African countries, and these have the potential to exacerbate child malnutrition and mortality.[50 51] Like in other settings,[52–55] we found evidence suggesting decreased childhood immunisation during the lockdown but the extent is unclear as healthcare providers reported using outreach services to vaccinate defaulters.

Healthcare services being considered as high-risk settings for infection influenced care seeking practices for children. Similar to reports in Nigeria and elsewhere, caregivers were avoiding hospitals for fear of contracting COVID-19.[49 56–58] The resultant self-management of childhood illness and decreased healthcare service utilisation are in keeping with other studies from Europe and Africa.[57–60] Studies within and outside Nigeria have also reported increased self-medication practice for the prevention and treatment of COVID-19-related symptoms but did not focus on self-medication for children during the pandemic.[61–63] A study conducted in Uganda also found higher neonatal mortality and morbidity during the lockdown.[64] Estimating the impacts of reduced hospital visits, seeking care from alternative sources, delayed hospital visits and increased self-medication for sick children was outside the scope of this study but will be crucial for understanding the indirect effects of COVID-19 public health measures. Nevertheless, our study supports the need for intelligent health communication and flexible approaches to increasing service delivery capacity, such as mobile outreach clinics to maintain healthcare access for children.[20 65] A study conducted in the UK hypothesised that decreased incidence of childhood illness during the lockdown period contributed to low paediatric admission for common and severe childhood illness during the lockdown;[66] however, hospital avoidance, care seeking from alternative sources and delayed presentation should not be dismissed.

The underlying distrust in government influenced COVID-19 perceptions, and provided the platform for the growing misinformation about the pandemic and this in turn shaped vaccine hesitancy.[67 68] Our findings are in agreement with studies in Nigeria which found that non-adherence to recommended preventive measures for COVID-19 was centred on political distrust, stemming from decades of perceived bad governance.[68 69] The mixed perception towards COVID-19 in Nigeria was therefore not surprising and similar controversies have been reported across several regions globally.[70] In times of uncertainty, a coping strategy is to use religion to provide explanations for strange events,[71] and these may conflict with emerging scientific evidence (particularly as conclusions change with new data) and frustrate containment measures.[72] Our findings support the need for inclusive risk communication for epidemic preparedness and control. Moreover, intervention adaptation to suit local contexts is essential during emergency response to epidemics.[45] Early reported cases of COVID-19 in the country were among foreigners and high-profile politicians. Linking COVID-19 results to known public officers could have been responsible for the perception that COVID-19 is a disease of the elite. In addition, limited testing capacity could have driven the perception that COVID-19 is not real, as up to 80% of infected individuals had been reported as mild or asymptomatic.[73]

Interestingly, the demand for COVID-19 vaccine was reportedly higher than anticipated among community members despite negative media reports and conspiracy theories. This finding is consistent with a study conducted by Arce *et al* which found higher willingness to receive COVID-19 vaccine in low and middle-income countries compared with high-income countries in which the survey was done.[74] Our findings support the call for vaccine equity, the need for sustained global partnership and continuous post-vaccination surveillance to achieve effective global vaccination for COVID-19.[75] The concern about the unprecedented short period to vaccine production and licensing underscores the need for sustained and increased efforts towards control of other communicable diseases like tuberculosis, HIV/AIDS and pneumonia— not neglecting other diseases because of COVID-19. Considering the background mistrust in government, donation of substandard vaccines and vaccines with short expiry dates or not valid for travel as well as conditional donation of vaccines feeds into public narratives of lack of trust in COVID-19 vaccines and reinforces conspiracy theories about COVID-19.[76–78] Meanwhile, vaccine hesitancy among healthcare providers requires attention for increased and sustained COVID-19 vaccine coverage in the long term.[79]

This study had limitations. First, we recruited caregivers from PHCs only and did not gather perspectives from other community members. This may mean that the perspectives captured here underestimate negative effects on care seeking. More so, given that participants were not consulted in the design of the interview guide, we acknowledge that finding from this study may not reflect all aspects considered important to the participants. Review of facility data shows a considerable decrease in outpatient attendance for children (online supplemental appendix V). Our findings have provided context-specific understanding of the indirect and direct effects of COVID-related public health measures and may inform future public health responses to disease outbreaks. Though the implementation of lockdown is context specific, findings from our study may be transferrable to other low and middle-income countries with a similar weak health system and where distrust of government has been a problem.

## CONCLUSION

The interpretation of the emergence of a new disease classification is dynamic and multifaceted. The COVID-19 pandemic in Lagos had both direct and indirect effects on care seeking for children. It is plausible that these had negative impacts on morbidity and mortality. Subsequent disease outbreak response requires active management of misinformation and intelligent health communication, including context-specific understanding of social media messaging and the role of religious institutions. Strengthening health and social support system interventions, notably around ensuring access to healthcare is not negatively affected, is crucial to building adaptive capacity for future disease outbreaks, pandemics and building public trust.

**Author affiliations**
[1]Department of Global Public Health, Karolinska Institutet, Stockholm, Sweden
[2]Department of Community Medicine, University College Hospital, Ibadan, Nigeria
[3]Department of Paediatrics, University College Hospital, Ibadan, Nigeria
[4]Centre for International Child Health, Murdoch Children's Research Institute, Royal Children's Hospital, University of Melbourne, Parkville, Victoria, Australia
[5]Department of Community Medicine, University of Ibadan College of Medicine, Ibadan, Nigeria
[6]Institute for Global Health, University College London, London, UK
[7]Department of Paediatrics, University of Ibadan College of Medicine, Ibadan, Nigeria

**Acknowledgements** We thank the clinical data collectors and facility heads for their support during the data collection and the caregivers and healthcare workers for giving us their time.

**Collaborators** Full INSPIRING Consortium: Carina King (Karolinska Institutet); Tim Colbourn, Rochelle Ann Burgess, Agnese Iuliano (UCL); Hamish Graham (Melbourne); Eric McCollum (Johns Hopkins); Tahlil Ahmed, Samy Ahmar, Christine Cassar, Paula Valentine (Save the Children UK); Adamu Isah, Adams Osebi, Magama Abdullahi, Ibrahim Haruna (Save the Children Nigeria); Temitayo Folorunso Olowookere (GSK Nigeria); Matt MacCalla (GSK UK); Adegoke Gbadegesin Falade, Ayobami Adebayo Bakare, Obioma Uchendu, Julius Salako, Funmilayo Shittu, Damola Bakare, Omotayo Olojede (University of Ibadan).

**Contributors** AAB, OEO, CK and HMA conceived the study. TC, CK and AGF are grant holders. AAB designed the study. OEO collected the data with oversight from AAB and OU. AAB and OEO led the analysis, with support from HMA, CK and HG. The manuscript was drafted by AAB with support from OEO, CK and HMA. All authors contributed to revisions and approved the final manuscript. CK is the guarantor of this study.

**Funding** This work was funded through the GlaxoSmithKline (GSK)-Save the Children Partnership (grant reference: 82603743).

**Disclaimer** Employees of both GSK and Save the Children UK contributed to the design and oversight of the wider INSPIRING study as part of a co-design process but did not take part directly in this substudy. Any views or opinions presented are solely those of the authors/publisher and do not necessarily represent those of Save the Children UK or GSK, unless otherwise specifically stated.

**Competing interests** SA, TA, CC and PV are employed by Save the Children UK who are part of the partnership funding the research. TFO and MM are employees of GSK, a multinational for-profit pharmaceutical company that produces pharmaceutical products for childhood pneumonia, including a SARS-CoV-2 vaccine, and no direct financial interests in oxygen or pulse oximeter products.

**Patient and public involvement** Patients and/or the public were involved in the design, or conduct, or reporting, or dissemination plans of this research. Refer to the Methods section for further details.

**Patient consent for publication** Parental/guardian consent obtained.

**Ethics approval** We obtained ethical approvals from the following ethics committees: Lagos State Primary Health Care Board (ref: LS/PHCB/MS/1128/VOL. V1/005), University of Ibadan/University College Hospital (ref: UI/EC/19/0551) and the University College London (ref: 3433/005). We obtained informed oral consent from all the participants and conducted the interviews under strict adherence to the study COVID-19 prevention protocol.

**Provenance and peer review** Not commissioned; externally peer reviewed.

**Data availability statement** Data are available upon reasonable request.

**ORCID iDs**
Ayobami Adebayo Bakare http://orcid.org/0000-0003-2456-7899
Carina King http://orcid.org/0000-0002-6885-6716
Hamish Graham http://orcid.org/0000-0003-2461-0463
Helle Molsted Alvesson http://orcid.org/0000-0001-6109-7203

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
