## [Reviewer comments · BMJ Open]

ARTICLE DETAILS

TITLE (PROVISIONAL)	Care seeking for under-five children and vaccine perceptions during the first two waves of the COVID-19 pandemic in Lagos State, Nigeria: a qualitative exploratory study
AUTHORS	Bakare, Ayobami; Olojede, Omotayo; King, Carina; Graham, Hamish; Uchendu, Obioma; Colbourn, Timothy; Falade, Adegoke; Alvesson, Helle

VERSION 1 – REVIEW

REVIEWER	Hoehl, Sebastian University Hospital Frankfurt, Institut of Medical Virology
REVIEW RETURNED	07-Dec-2022

GENERAL COMMENTS	To better uphold child health in future pandemics, data, especially from low-and-middle-income countries, who “share a higher burden of childhood morbidity and mortality”, is required. This study aims to understand care-seeking practices and ambiguity towards vaccine uptake for children below the age of five years during the first two waves of the COVID pandemic in Lagos, Nigeria in an exploratory qualitative study using semi-structured interviews with caregivers and healthcare providers. A thematic analysis was conducted. The results indicate that care seeking for young children was affected in Lagos, Nigeria, and vaccine hesitancy in connection with distrust in the government and misinformation a concern. The manuscript is very well written. The methodology is presented clearly, and the study’s limitations are sufficiently noted. The date on literature reference one should be checked. I have no further remarks and thank the authors for letting me review their work.
---

REVIEWER	Neill, Sarah University of Plymouth, School of Nursing and Midwifery
REVIEW RETURNED	16-Dec-2022

GENERAL COMMENTS	Thank you for submitting this paper. It was very interesting to read about parents help seeking during the pandemic in Nigeria. I have identified a few things which could make your paper stronger:
--

	1. A careful review of the use of English - there are some unusual phraseology which make some thing less easy to understand than could be the care. 2. Make it clear in your objectives that you were asking specifically about vaccine decision making not just care-seeking. 3. I would have liked to score study design as somewhat appropriate. A qualitative approach was appropriate but the highly structured interview schedules will have focussed the data collected only on those areas the study team deemed of most interest, meaning that the data does not necessarily reflect what was of most importance to your participants. This may be the results of not including caregivers in the co-design. This needs to be recognised as a limitation of the paper. 4. It is not always clear when the data reported comes from parents/care-givers and when it comes from healthcare providers. I suggest that the findings (and you should use the term Findings for a qualitative research report, not Results) are restructured to more clearly show which group of respondents the findings are from. 5. It was good to see a table of themes however there seems to be some mixing of methodologies here as you use themes and categories. Themes and categories are different. Please review thematic analysis methodology literature and correct your terminology accordingly. You may find that you can condense some of your subdivisions which would help with clarity of the findings presented. Some of your quotes could also be shorter. It is acceptable to remove parts of quotes which are not pertinent to the findings it is used to support.
--	---

VERSION 1 – AUTHOR RESPONSE

Reviewer: 1 - Dr. Sebastian Hoehl, University Hospital Frankfurt

To better uphold child health in future pandemics, data, especially from low-and-middle-income countries, who “share a higher burden of childhood morbidity and mortality”, is required. This study aims to understand care-seeking practices and ambiguity towards vaccine uptake for children below the age of five years during the first two waves of the COVID pandemic in Lagos, Nigeria in an exploratory qualitative study using semi-structured interviews with caregivers and healthcare providers. A thematic analysis was conducted. The results indicate that care seeking for young children was affected in Lagos, Nigeria, and vaccine hesitancy in connection with distrust in the government and misinformation a concern. The manuscript is very well written. The methodology is presented clearly, and the study’s limitations are sufficiently noted.

Response: Thankyou for this summary of our paper, and supportive comments!

The date on literature reference one should be checked.

Response: corrected

Reviewer: 2 - Prof. Sarah Neill, University of Plymouth

Thank you for submitting this paper. It was very interesting to read about parents help seeking during the pandemic in Nigeria. I have identified a few things which could make your paper stronger:
Response: We appreciate the comments given and believe our revisions have strengthened the paper.

1. A careful review of the use of English - there are some unusual phraseology which make some thing less easy to understand than could be the care.

Response: Thank you for pointing out this. Several of the authors are native English speakers and have re-read the manuscript and made edits for readability.

2. Make it clear in your objectives that you were asking specifically about vaccine decision making not just care-seeking.

Response: vaccine decision making has been explicitly stated in the objective (page 2 line 5 and page 4 lines 40-45).

3. I would have liked to score study design as somewhat appropriate. A qualitative approach was appropriate but the highly structured interview schedules will have focussed the data collected only on those areas the study team deemed of most interest, meaning that the data does not necessarily reflect what was of most importance to your participants. This may be the results of not including caregivers in the co-design. This needs to be recognised as a limitation of the paper.

Response: thank you for your suggestion. The section on limitation has been revised to incorporate your suggestion.

“This study has some limitations. We recruited caregivers from PHCs only and did not gather perspectives from other community members. This may mean that the perspectives captured here underestimates negative effects on care-seeking. More so, given that participants were not consulted in the design of the interview guide, we acknowledge that findings from this study may not reflect all aspects considered important by the participants.”

4. It is not always clear when the data reported comes from parents/caregivers and when it comes from healthcare providers. I suggest that the findings (and you should use the term Findings for a qualitative research report, not Results) are restructured to more clearly show which group of respondents the findings are from.

Response: We have added clarifications throughout the findings section to indicate when the themes and subthemes are not shared by both caregivers and healthcare providers, and added a statement in the methods section to indicate this (page 7, line 23).

5. It was good to see a table of themes however there seems to be some mixing of methodologies here as you use themes and categories. Themes and categories are different. Please review thematic analysis methodology literature and correct your terminology accordingly. You may find that you can condense some of your subdivisions which would help with clarity of the findings presented. Some of your quotes could also be shorter. It is acceptable to remove parts of quotes which are not pertinent to the findings it is used to support.

Response: We have reduced the length of some of the quotes, so they are more focused. For Table 2, we have re-worded the headings to correctly reflect the thematic approach. We used organizing themes, themes and sub-themes, and removed the final column for clarity.

VERSION 2 – REVIEW

REVIEWER	Neill, Sarah University of Plymouth, School of Nursing and Midwifery
REVIEW RETURNED	15-Feb-2023

GENERAL COMMENTS	The paper is much improved following the revisions. This paper will make an important contribution to research concerning the impact of Covid-19 in Africa. Well done! There are two very small things to correct - carefully read the table of lockdown restrictions - there are some typos and inconsistent use of terms (e.g. open or opening). At the end of the abstract there is one misleading statement 'containing misinformation is crucial' sounds as if you are advocating misinformation when I think you mean that misinformation must be corrected.
---

VERSION 2 – AUTHOR RESPONSE

Care seeking for under-five children and vaccine perceptions during the first two waves of the COVID-19 pandemic in Lagos State, Nigeria: a qualitative exploratory study

We thank the reviewers for the helpful feedback.

Reviewer: 2

Prof. Sarah Neill, University of Plymouth Comments to the Author:

The paper is much improved following the revisions. This paper will make an important contribution to research concerning the impact of Covid-19 in Africa. Well done!

Response: Thank you for your useful comments which helped us to improve the manuscript.

There are two very small things to correct - carefully read the table of lockdown restrictions - there are some typos and inconsistent use of terms (e.g. open or opening).

Response: Thank you for pointing out this. Necessary corrections have been made.

At the end of the abstract there is one misleading statement 'containing misinformation is crucial' sounds as if you are advocating misinformation when I think you mean that misinformation must be corrected.

Response: Thank you for pointing out this. The statement has been rephrased to “Strengthening health and social support systems with context-specific interventions and correcting misinformation is crucial to building adaptive capacity for response to future pandemics.”